Genetic differentiation and phylogeography of Mediterranean-North Eastern Atlantic blue shark (Prionace glauca, L. 1758) using mitochondrial DNA: panmixia or complex stock structure?

Leone Agostino agostino.leone2@unibo.it 1
Urso Ilenia 1
Damalas Dimitrios 2 12
Martinsohn Jann 2
Zanzi Antonella 2
Mariani Stefano 3
Sperone Emilio 4
Micarelli Primo 5
Garibaldi Fulvio 6
Megalofonou Persefoni 7
Bargelloni Luca 8
Franch Rafaella 8
Macias David 9
Prodöhl Paulo 10
Fitzpatrick Séan 10
Stagioni Marco 11
Tinti Fausto 1
Cariani Alessia 1
1 Department of Biological, Geological and Environmental Sciences (BiGeA), Laboratory of Genetics & Genomics of Marine Resources and Environment (GenoDREAM), University of Bologna , Ravenna , Italy
2 Joint Research Centre (JRC), Directorate D-Sustainable Resources, Unit D2-Water and Marine Resources, European Commission , Ispra , Italy
3 School of Environment & Life Sciences, University of Salford , Salford, Greater Manchester , United Kingdom
4 Department of Biology, Ecology and Earth Science (DiBEST), University of Calabria , Arcavacata di Rende , Italy
5 Sharks Research Center, Aquarium Mondo Marino , Massa Marittima , Italy
6 Department of Earth, Environment and Life Sciences, University of Genova , Genova , Italy
7 Department of Biology, National and Kapodistrian University of Athens , Ilissia , Greece
8 Department of Comparative Biomedicine and Food Science, University of Padova , Legnaro , Italy
9 Centro Oceanográfico de Málaga, Instituto Español de Oceanografía , Málaga , Spain
10 School of Biological Sciences, Queen’s University, Institute for Global Food Security , Belfast , United Kingdom
11 Department of Biological, Geological and Environmental Sciences, Marine Biology and Fisheries Laboratory, University of Bologna , Fano , Italy
12 Current affiliation:  Hellenic Centre for Marine Research , Athinon , Anavyssos , Greece
Castilho Rita
Electronic publication date: 2017 Dec 6
Publication date: 2017
Volume: 5
Electronic Location ID: e4112
Received 2017 Aug 21; Accepted 2017 Nov 9
Copyright: ©2017 Leone et al.
Copyright year: 2017
Copyright holder: Leone et al.
License: This is an open access article distributed under the terms of the Creative Commons Attribution License, which permits unrestricted use, distribution, reproduction and adaptation in any medium and for any purpose provided that it is properly attributed. For attribution, the original author(s), title, publication source (PeerJ) and either DOI or URL of the article must be cited.
License URL: https://creativecommons.org/licenses/by/4.0/

Keywords: mtDNA, Population expansion, Blue shark, Phylogeography, Geographical breaks, Mediterranean stocks, Sharks

Funding: European Commission, Joint Research Center (JRC) University of Bologna, Italy This research was funded by the European Commission, Joint Research Center (JRC) with the Tender “MEDBLUESGEN—A population genetic study on Mediterranean blue shark for stock identification and conservation” (https://fishreg.jrc.ec.europa.eu/web/medbluesgen/; Negotiated call Procedure no. JRC/IPR/2014/G.3/0023/NC and Contract number 259844), and by the Genetics & Genomics of Marine Resources and Environment Laboratory (GenoDREAM), Department of Biological, Geological & Environmental Sciences (BiGeA) of the University of Bologna, Italy. The funders had no role in study design, data collection and analysis, decision to publish, or preparation of the manuscript.

==============================
Background

The blue shark (Prionace glauca, Linnaeus 1758) is one of the most abundant epipelagic shark inhabiting all the oceans except the poles, including the Mediterranean Sea, but its genetic structure has not been confirmed at basin and interoceanic distances. Past tagging programs in the Atlantic Ocean failed to find evidence of migration of blue sharks between the Mediterranean and the adjacent Atlantic, despite the extreme vagility of the species. Although the high rate of by-catch in the Mediterranean basin, to date no genetic study on Mediterranean blue shark was carried out, which constitutes a significant knowledge gap, considering that this population is classified as “Critically Endangered”, unlike its open-ocean counterpart.

Methods

Blue shark phylogeography and demography in the Mediterranean Sea and North-Eastern Atlantic Ocean were inferred using two mitochondrial genes (Cytb and control region) amplified from 207 and 170 individuals respectively, collected from six localities across the Mediterranean and two from the North-Eastern Atlantic.

Results

Although no obvious pattern of geographical differentiation was apparent from the haplotype network, Φst analyses indicated significant genetic structure among four geographical groups. Demographic analyses suggest that these populations have experienced a constant population expansion in the last 0.4–0.1 million of years.

Discussion

The weak, but significant, differences in Mediterranean and adjacent North-eastern Atlantic blue sharks revealed a complex phylogeographic structure, which appears to reject the assumption of panmixia across the study area, but also supports a certain degree of population connectivity across the Strait of Gibraltar, despite the lack of evidence of migratory movements observed by tagging data. Analyses of spatial genetic structure in relation to sex-ratio and size could indicate some level of sex/stage biased migratory behaviour.

Introduction

The blue shark (Prionace glauca, Linnaeus 1758; BS henceforth) is one of the most abundant epipelagic sharks that is found in all oceans from 60°N to 50°S (Compagno, 1984). Blue sharks are rarely targeted by commercial fishing, but feature prominently as by-catch of fisheries targeting large pelagic fish, especially swordfish and tuna longlines (Fowler et al., 2005). BS populations trend data are available only for a part of the geographic range and stock assessments are highly uncertain (Dulvy et al., 2014; Coelho et al., 2017); due to the huge amount of by-caught BS (approx. 20 million per annum, Stevens, 2009), the species has being categorized worldwide as “Near Threatened” in the IUCN Red List (Stevens, 2009). Based on recent assessment (ICCAT, 2015), the North Atlantic stock is unlikely to be currently overfished. The Mediterranean BS, on the other hand, is estimated to have undergone a 90% decline over three generations, primarily due to overfishing (Ferretti et al., 2008), and is now categorized as “Critically Endangered” (Sims et al., 2016). Given the vast amount of poorly reported by-catch, the increasing commercial value of the species (Megalofonou, Damalas & Yannopolous, 2005) and the persistent issue of the global trade in shark fin products, of which BS is the main component (Clarke et al., 2006), a more explicit management is needed for this species, which should be underpinned by robust knowledge of its population structure.

In the Atlantic, BS is distributed from Canada to Argentina, on the western side, and from Norway to South Africa on the eastern side, including the Mediterranean Sea (Compagno, 1984). The population structure and dynamics of Atlantic BS is still poorly known, despite several long-term tagging studies, which revealed extensive movements of BS tagged in the western side of the North Atlantic (henceforth NA), with well documented eastward trans-Atlantic migrations (Kohler, Casey & Turner, 1998; Kohler et al., 2002; Kohler & Turner, 2008; Vandeperre et al., 2014). Sexual segregation was also evident, with a concentration of mature females in more temperate waters of the northernmost NA, and immature males predominant in the southernmost NA (Sampaio da Costa, 2013). Mature BS of both sexes seemed to be distributed in the southern part of NA, while immature individuals of both sexes and sub-adult females are usually distributed in the northern areas (Kohler et al., 2002). Conversely, a prevalent occurrence of immature juveniles is reported in the Mediterranean Sea (Megalofonou, Damalas & De Metrio, 2009; Kohler et al., 2002). A significant genetic heterogeneity among potential BS nurseries from the Atlantic Ocean (Portugal and Azores) and those from South Africa was detected by Sampaio da Costa (2013) from mitochondrial and nuclear marker variation. Their finding indicated a deeper separation between the northern and the southern NA nurseries and supported a male philopatry behaviour to mating areas exclusively contributing to a single nursery ground. Contradictorily, a recent genetic survey (Veríssimo et al., 2017) carried out on the same dataset (i.e., young-of-year and <2 years juveniles) collected from the same nurseries, enriched with more samples from different areas (i.e., coasts of Brazil), and using the same type of markers, showed a lack of spatio-temporal genetic differentiation, suggesting the presence of a panmictic population in the whole Atlantic.

To date, no genetic data are available for the Mediterranean BS population and population structure and dynamics of BS in the Mediterranean are presently inferred only by Atlantic-Mediterranean integrated tagging studies and fishing data assessments (Kohler, Casey & Turner, 1998; Kohler et al., 2002; Ferretti et al., 2008; Kohler & Turner, 2008; Megalofonou, Damalas & De Metrio, 2009).

Irrespective of the small recapture rate (out of the 91,450 BS specimens tagged in the north western Atlantic, only 5.9% were recaptured), extensive tag-recapture surveys carried out from 1962 to 2000, indicated that North Atlantic BS form a single stock and that trans-Atlantic migratory movements were quite frequent, likely favoured by the oceanic current system (Kohler et al., 2002). Focusing on the Atlantic–Mediterranean connectivity, the reproductive migratory movements of Atlantic BS towards Mediterranean and the degree of population connectivity between the two areas are still unknown, because only one adult BS male tagged in the north-western Atlantic and one sub adult female tagged in the North-Eastern Atlantic were recaptured in the Mediterranean (Kohler et al., 2002). The large majority of BS tagged in the Mediterranean Sea were immature and remained in the tagging area, with the only exception of a subadult female that moved a short distance to the adjacent north-eastern Atlantic area. Most of the BS caught in the Mediterranean (99% and 98% for males and females, respectively) are immature, indicating that the Mediterranean BS stock consists primarily of small immature BS of both sexes, with a sex-ratio skewed toward females or males, depending on different geographical areas (Kohler et al., 2002; Megalofonou, Damalas & De Metrio, 2009). A high number of pregnant females was observed in the Adriatic, North Ionian Sea and Ligurian Sea, suggesting potential nursery grounds for BS (Megalofonou, Damalas & De Metrio, 2009; F Garibaldi, pers. comm., 2017). On the other hand, the adjacent South-Eastern North Atlantic BS was prevalently composed by primarily mature individuals of both sexes with male-based sex ratio.

The primary aims of this study is to test the null hypothesis of panmixia between North Atlantic and Mediterranean BS, by comparing the mtDNA genetic variation of two gene regions, the control region (CR) and the Cytochrome b (Cytb) among four population samples collected in the North-Eastern and South-Eastern North Atlantic and in the Western and Eastern Mediterranean. Given the female philopatry observed in other carcharhiniformes (Mourier & Planes, 2013; Tillett et al., 2012), mtDNA markers are likely to be useful to spot localised groups due to site-fidelity. Accordingly, this work aims to provide further and needed data on matrilineal genetic structure, female philopatry and demography of Mediterranean BS. These, previously lacking, data will contribute to a better understanding and inclusion of the Mediterranean BS dynamics in the wider North Atlantic population model, to improve assessment and management of BS stocks in the area.

Material and Methods

Blue shark sampling

Sampling was carried out between 2003 and 2016 and tissue specimens and individual data were collected by means of commercial fishermen and scientific surveys. Mediterranean BS were collected from multiple locations in the Eastern (Central Adriatic, CADR, 21; Ionian Sea, IONI, 15; Aegean Sea and Levantine Sea, AEGE, 20) and Western areas (South Tyrrhenian, TYRR, 10; Ligurian Sea, LIGU, 57; Balearic Islands, BALE, 42). North Atlantic BS were caught from the North Eastern Atlantic Ocean off the coasts of Portugal (SNEATL, 33) and Celtic Sea (NNEATL, 16) (Fig. 1). A total of 214 BS individuals were collected (N = 91 males, N = 101 females and N = 22 unsexed) (Table S1). The BS individuals were grouped according the Total Length (TL) in three size categories (Pratt, 1979; Vandeperre et al., 2014): juveniles (J, TL ≤ 120 cm), young (Y, TL = 120–180 cm) and large (L, TL ≥ 180 cm).

Figure 1 Sampling sites of Mediterranean and North Eastern Atlantic Blue Sharks.

North North-Eastern Atlantic (NNEATL, red dots, N = 16), South North-Eastern Atlantic (SNEATL, blue dots, N = 33), Western Mediterranean (WMED, purple dots, N = 109) and Eastern Mediterranean (EMED, green dots, N = 56). The map was created using R version 3.4.1 (R Core Team, 2016; Becker, Wilks & Brownrigg, 2017).

A unique and transparent sampling documentation tool was developed within the project, in order to render data public. This tool can be used by everyone as an interactive map visiting the website: https://fishreg.jrc.ec.europa.eu/web/medbluesgen/sampling-data.

Molecular methods

Individual fin clips or skeletal muscle tissue samples were collected and preserved in 96% ethanol and kept at −20 °C until laboratory analyses. DNA extraction was carried out using the Invisorb® Spin Tissue Kit, Invitek (STRATEC Molecular, Birkenfeld, Germany) and the Wizard® Genomic DNA Purification Kit (Promega, Madison, WI, USA) following the manufacturers’ protocols.

Species-specific primer pairs for the amplification of the mitochondrial control region (CR) and cytochrome b (Cytb) genes were designed. Homologous complete CR and Cytb sequences of Prionace glauca available in GenBank were retrieved and aligned using ClusterW algorithm implemented in MEGA ver.7.0 (Tamura et al., 2013). Primer pairs were designed using the online software PRIMER3 (ver.0.4.0) (Untergasser et al., 2012), minimizing the propensity of oligos to form hairpins or dimers or to hybridize or prime from unintended sites in the full mitochondrial BS genome (Acc. Num. NC_022819, Chen et al., 2013).

The designed primer pairs (control region: CR-Blues-F 5′AAACACATCAGGGGAAGGA G3′, CR-Blues-R 5′CATCTTAGCATCTTCAGTGCC3′; Cytochrome-b: Cytb-Blues-F 5′TCCTCACAGGACTCTTCCTAGC3′, Cytb-Blues-R 5′GTCGAAAGATGGTGCTTCGT3′) were tested using a temperature gradient to identify the most suitable melting temperatures (Tm = from 50 °C to 60 °C) according to PCR cycling conditions described by Ovenden et al. (2009).

Once the optimal melting temperature was identified, the PCR thermal profile was adjusted and the PCR reactions were performed for both markers in a final volume of 50 µL containing 31.75 µL of distilled sterile H2O, 8 µL of Buffer 10× (Tris-HCl; final 1×), 3 µL of MgCl2 (25 mM; final 1.5 mM), 2 µL of dNTPs (10 mM; final 0.37 mM), 2.5 µL (10 µM; final 0.46 µM) of each primer, 0.25 µL (5U/µL; final 1.5U) of Taq polymerase and 2 µL of template DNA(10–20 ng). The temperature profile included an initial denaturation at 94 °C for 2 min, followed by 35 cycles of denaturation at 94 °C for 30 s, annealing at 60 °C for 30 s, elongation at 72 °C for 30 s and a final elongation step at 72 °C for 5 min. PCR amplicons were sequenced using the external service provider MACROGEN® Europe.

Data analysis

The CR and Cytb nucleotide sequences obtained were validated with the homologous gene sequences deposited in the GenBank with the BLASTn search implemented in the NCBI website (Altschul et al., 1990), and aligned using the ClusterW algorithm implemented in MEGA ver.7.0 (Tamura et al., 2013). When aligned to the complete BS mitochondrial genome, Cytb sequences mapped from nucleotide position 14,530 to 15,291 and CR from 15,651 to 16,397.

Given the high potential of geographical dispersal of the species, sequence data were grouped according to the four geographical areas: EMED, WMED, SNEATL and NNEATL (Fig. 1). The software DNAsp v.5.10.01 (Librado & Rozas, 2009) was used to assess the genetic diversity parameters at both markers: the number of haplotypes (Nh), the number of polymorphic sites (S), the haplotype (h) and nucleotide diversity (π) with associated standard deviation (stdev).

Haplotype relationships were inferred using the dnaml program of the PHYLIP package version 3.6 (Felsenstein, 1989; Felsenstein, 2005) implemented in the software program HaploViewer (http://www.cibiv.at/ greg/haploviewer).

Partition of molecular variance and its significance was estimated with the AMOVA (Excoffier, Smouse & Quattro, 1992) implemented in Arlequin ver 3.5.2.2 software (Excoffier & Lischer, 2010), testing four alternative groupings of geographical sampling locations (1: no groups; 2: NNEATL+SNEATL vs WMED+EMED; 3: NNEATL+SNEATL vs WMED vs EMED; 4: NNEATL vs SNEATL vs WMED vs EMED). Haplotype frequencies and pairwise ΦST with the associated p-values were calculated using the software Arlequin ver 3.5.2.2 (Excoffier & Lischer, 2010) after 20,000 permutations, setting up a α = 0.05 significance threshold level.

Demographic history was investigated using the mismatch distribution as implemented in the DNAsp software (Librado & Rozas, 2009).

Furthermore, historical demographic trend of the four groups was investigated using Bayesian Skyline Plot (BSP) analysis implemented in the software BEAST v.1.8.2 (Drummond et al., 2005; Drummond et al., 2012), using the best evolutionary models for both Cytb and CR markers inferred using JModelTest 2.1.1 (Darriba et al., 2012), and the average mutation rate for sharks, 0.62% and 0.31% for CR and Cytb respectively (Martin & Palumbi, 1993; Galván-Tirado et al., 2013). The same software and parameters, with associate software TreeAnnotator and FigTree, were used to define the phylogeny of the Mediterranean and Eastern Atlantic BS populations.

Results

Among sexed individuals (N = 192; Table S1), BS females significantly outnumbered males in the NA samples (sex-ratio 0.34, χ2 test: 10.256 P2tail = 0.001; P1tail = 0, d.f. 1) while in the two Mediterranean BS groups a weak and not significant predominance of males was observed (WMED: 1.19, χ2 test: 0.786 P2tail = 0.375; P1tail = 0.188, d.f. 1; EMED: 1.09, χ2 test: 0.087 P2tail = 0.768; P1tail = 0.384, d.f. 1). Sized BS (N = 209) were composed by 63 juvenile, 82 young and 64 large individuals (Table S1). In the NA and WMED the young BS (TL = 120–180 cm; 48% and 50%, respectively) were predominant, while in the EMED a large predominance of juveniles was observed (TL ≤ 120 cm; 63%). Noticeably 67% of the BS sampled in the Ionian Sea and 95% of those sampled in the Adriatic Sea were juveniles. Large BS are similarly represented in the geographical groups with percentages varying from 25% (EMED) to 34% (NA), full details presented in Table S1.

A total of 207 and 170 BS individuals were sequenced for Cytb (762 bp) and CR (747 bp), respectively. Haplotype sequences (Cytb, N = 23 and CR, N = 55) were deposited in GenBank under the accession numbers MG515900–MG516106 and MG545732–MG545901 for Cytb and control region, respectively.

The Cytb sequence dataset exhibited 16 polymorphic segregating sites while CR dataset showed 27 polymorphic segregating sites. The Cytb haplotype diversity ranged from 0.784 to 0.835, and that of the CR from 0.894 to 1.000. The Cytb nucleotide diversity ranged from 0.001 to 0.002, and that of the CR from 0.004 to 0.008. Detailed genetic diversity of BS samples collected from the four macro areas and all sampling locations is presented in Table 1 and Table S2, respectively.

Table 1 Mitochondrial gene polymorphism of Prionace glauca population samples subdivided according to the four macro areas.

POP	N	Nh	S	h	stdev h	π	stdevπ	
Cytb	
NNEATL	14	9	6	0.835	0.010	0.00231	0.00046	
SNEATL	33	8	10	0.822	0.034	0.00200	0.00038	
WMED	105	13	6	0.801	0.023	0.00167	0.00011	
EMED	55	10	6	0.784	0.033	0.00151	0.00013	
TOTAL	207	23	16	0.821	0.013	0.00184	0.00010	
CR	
NNEATL	6	6	15	1.000	0.093	0.00812	0.00106	
SNEATL	33	17	13	0.932	0.026	0.00424	0.00038	
WMED	79	34	18	0.949	0.011	0.00418	0.00019	
EMED	52	19	12	0.894	0.028	0.00382	0.00031	
TOTAL	170	55	27	0.951	0.006	0.00453	0.00014	
Notes.

N number of individuals

Nh number of haplotypes

S Number of segregating informative sites

h haplotype diversity and associate standard deviation

π nucleotide diversity and associate standard deviation

NNEATL North North–eastern Atlantic

SNEATL South Northeastern Atlantic

WMED Western Mediterranean

EMED Eastern Mediterranean

The Cytb and CR haplotype networks highlighted the distribution of haplotypes irrespective of the geographical origin of BS samples, indicating the lack of phylogeographical structure in the Mediterranean and adjacent North Atlantic BS (see Fig. 2, Fig. S1). In the Cytb network, the four main frequent haplotypes were shared by BS from all the four geographical areas, except for the most frequent haplotype which was shared by BS from the three geographical areas, SNEATL, WMED and EMED. In the CR network, six most frequent haplotypes (No. individuals ≥ 10) were observed. Although these six haplotypes were shared by all geographical areas, three of them were shared by Mediterranean and SNEATL, one by Mediterranean and NNEATL, and two within the Mediterranean. In both networks, most of the NNEATL haplotypes were singletons (Fig. 2).

Figure 2 Cytochrome-b (A) and Control Region (B) Maximum Likelihood Haplotype Network of Mediterranean/North East Atlantic Blue Shark collected from the four geographical areas.

NNEATL: North North–Eastern Atlantic; SNEATL: South North–Eastern Atlantic; WMED: Western Mediterranean; EMED: Eastern Mediterranean.

The AMOVA (Table 2) revealed a significant overall ΦST among population samples for both markers. Significant partition of molecular variance among areas was observed when BS sampling locations were grouped according to the four geographical areas in both markers (AMOVA4), according to three areas (NEATL (NNEATL+SNEATL) vs WMED vs EMED; AMOVA3), and according to two areas (NEATL (NNEATL+SNEATL) vs MED (WMED+EMED), for both dataset. However, the grouping that best described the partitioning of genetic variance is when the different sampling locations are subdivided into four areas showing the lowest partition of molecular variance among populations within group.

Table 2 Analysis of molecular variance (AMOVA) of Cytochrome b (Cytb) and Control Region (CR) of the Mediterranean and North–eastern Atlantic Blue Sharks (Prionace glauca).

	Cytb	CR	
	% variation	Φ-Statistics	p	% variation	Φ-Statistics	p	
AMOVA1: Overall (all population samples)	
Among populations	8.20			11.25			
Within populations	91.80	ST = 0.0819	0.00000	88.75	ST = 0.11249	0.00000	
AMOVA2: 2 groups: (NNEATL+SNEATL vs WMED+EMED)	
Among groups	12.39	CT = 0.1239	0.03496	7.89	CT = 0.0788	0.03471	
Among pops within group	2.40	SC = 0.0273	0.02287	7.41	SC = 0.0804	0.00005	
Within populations	92.68	ST = 0.1479	0.00000	84.70	ST = 0.1529	0.00000	
AMOVA3: 3 groups: (NNEATL+SNEATL vs WMED vs EMED)	
Among groups	7.01	CT = 0.0701	0.02188	5.68	CT = 0.0568	0.03656	
Among pops within group	2.84	SC = 0.0305	0.02397	6.78	SC = 0.0719	0.00075	
Within populations	90.15	ST = 0.0985	0.00000	87.54	ST = 0.1246	0.00000	
AMOVA4: 4 groups: (NNEATL vs SNEATL vs WMED vs EMED)	
Among groups	8.87	CT = 0.0887	0.02073	7.93	CT = 0.0793	0.03726	
Among pops within group	1.20	SC = 0.0132	0.13076	4.89	SC = 0.0531	0.00649	
Within populations	89.92	ST = 0.1007	0.00000	87.18	ST = 0.1282	0.00000	

With the Cytb sequence data, all pairwise ΦST values among the four geographical areas were significant except that between the two Atlantic groups (ΦST = 0.1152; p = 0.019) that became non-significant after the Bonferroni correction for multiple tests (Martin & Douglas, 1995) (Table 3). Unlike the CR dataset, only the pairwise ΦST values between SNEATL and the two Mediterranean areas and between WMED and EMED remained significant after the Bonferroni correction for multiple tests (Table 3).

Table 3 Pairwise Φst values (below the diagonal) and associated p-values (above the diagonal) among the blue sharks of the four geographical areas.

	NNEATL	SNEATL	WMED	EMED	
Cytb	
NNEATL		0.01868*	0.00000	0.00000	
SNEATL	0.08167*		0.00055	0.00015	
WMED	0.23969	0.08633		0.20052	
EMED	0.29481	0.12441	0.00658		
CR	
NNEATL		0.0097*	0.0482*	0.0187*	
SNEATL	0.1649*		0.0003	0.0000	
WMED	0.1061*	0.1049		0.0072	
EMED	0.1620*	0.2188	0.0463		
Notes.

* Values that resulted not significant after the Bonferroni correction for multiple tests (a-level of significance after Bonferonni correction: p = 0.0083).

AMOVA and pairwise ΦST analyses were performed on a reduced dataset, selecting only juvenile and immature specimens from each sampling site. Despite the reduced sample sizes and the complete absence of data from the site NNEATL, the results obtained are in agreement with the values observed with the complete dataset (Tables S3 and S4).

The Cytb distribution of sequence mismatch pairwise differences showed a skewed unimodal distribution in all four BS macro areas suggesting a recent bottleneck or sudden population expansion (Fig. S2). A unimodal mismatch distribution was obtained with CR dataset in the NNEATL BS. The CR mismatch distribution of EMED, SNEATL and NNEATL BS resulted to a slightly ragged pattern (Fig. S2) that could suggest a more constant population size of the Mediterranean BS over generations.

Both BSP analyses suggested a constant population size increase of Mediterranean and North–eastern Atlantic BS, starting more recently in the Mediterranean than in the North–eastern Atlantic (∼0.02–0.15 Mya vs 0.15–0.4 Mya; Fig. 3). Divergence time analysis based on both markers (Fig. S3) highlights a similar pattern of separation between two main groups, composed by BS from all regions, without any evidence of separation between defined geographic areas. The separation between the two clades, which is strongly supported of Posterior Probability (PP = 1.0) in both markers, is dated back to 1.24 Mya and 0.94 Mya using Cytb and control region, respectively.

Figure 3 Bayesian Skyline Plot from the Cytb, A–D, and control region, E–H, of the four different geographical areas.

NNEATL: North North–eastern Atlantic; SNEATL: South North–eastern Atlantic; WMED: Western Mediterranean; EMED: Eastern Mediterranean. The Y-axis indicates effective population size (Ne) × generation time, while the X-axis indicates mean time in million of years before present. The thick line represents the average, while the blue band represents 95% highest posterior density (HPD) intervals.

Discussion and Conclusions

The BS is probably the most mobile shark species in the world (Stevens, 1990) and past research works, using both mitochondrial and nuclear markers, have struggled to find genetic structure at interoceanic scale (Sampaio da Costa, 2013; King et al., 2015; Li et al., 2016; Veríssimo et al., 2017). This high level of gene flow make it difficult to define clear BS population units. In the Pacific Ocean, the lack of structure may be the result of the combination of high potential of migration and the lack of effective barriers to gene flow (Veríssimo et al., 2017).

Experimental data have indicated that no significant genetic structure is detected in spatially distant BS samples (King et al., 2015; Li et al., 2016; Veríssimo et al., 2017). Our results revealed significant signals of geographical structuring for Mediterranean and adjacent Atlantic BS, with several frequent mtDNA haplotypes that are exclusive of the Mediterranean BS and other that are shared with the Atlantic population samples.

While both haplotype networks failed to evidence a clear geographical structure, either between Mediterranean and North Atlantic BS or within the Mediterranean, the results of AMOVA revealed a significant partition of molecular variance among all population samples and when they were grouped according to the four geographical areas with both mitochondrial markers (8.87% for Cytb and 7.93% for CR). Previous studies carrying out AMOVA on the Atlantic BS using the control region variation, showed a significance variance among groups formed by the North Atlantic BS collected from Portugal and Azores and by the South African (See Table 7 of Sampaio da Costa, 2013) or Brazilian BS (Veríssimo et al., 2017). On the contrary, the global population genetics carried out by Fitzpatrick (2012), using concatenated fragments from: 16S, tRNA, COII, ATPase and control region genes, showed no significance variation among oceans, based upon comparisons between North Atlantic and all sampling locations combined (See Table 5.7 of Fitzpatrick, 2012). Although BS exhibits high potential of dispersal and migration, our results seem to reject an absence of geographical structure in the Mediterranean and adjacent North-eastern Atlantic BS. The pairwise Φst analysis revealed a geographical structuring between the two Mediterranean groups and Southern North–eastern Atlantic BS, with a closer genetic similarity of the Southern North–eastern Atlantic with the Western Mediterranean BS rather than with the Eastern Mediterranean BS. This pattern of differentiation seems to suggest that reproductive movements, such as female philopatry, may occur between the Western Mediterranean and the Southern North–eastern Atlantic BS. In addition, pairwise Φst values highlighted that the EMED BS are the more divergent from the NATL BS Given that SNNEATL specimens are from a previously identified nursery site (Veríssimo et al., 2017), the pairwise ΦST values could suggest that specimens from WMED can be reproductively related to the SNNEATL, while EMED could represent a nursery site in itself (Megalofonou, Damalas & De Metrio, 2009).

Our sampling work has also preliminarily revealed significant differences between North-eastern Atlantic and Mediterranean BS by sex-ratio and size. This pattern could be the result of a sex-biased reproductive migratory behaviour that could contribute to explain the significant phylogeographical structure. Similarly, size differences were observed between WMED and EMED BS, with the large and sexually mature individuals abundant in the easternmost Mediterranean sampling location (Aegean Sea) while the sub-adult and juvenile BS frequent in the Adriatic and Ionian Seas. The great abundance of juvenile BS in the Adriatic Sea seemed to confirm the nursery role of this area for BS (Megalofonou, Damalas & De Metrio, 2009). The biological data reported in Megalofonou and colleagues (2009) describe a larger amount of big female in the easternmost Mediterranean (e.g., Aegen Sea) which is in agreement with the pattern inferred from our dataset. Conversely, using data on size and maturity stages, Kohler and colleagues (2002) observe that the majority of sharks from the Mediterranean Sea were juvenile and immature (99% of males and 98% of females; mean = 65 cm of fork length). The difference may be related to the different sampling design and fishing gear used in the studies. In fact, the majority of the data collected by Kohler and colleagues (2002) came from volunteer recreational fishermen, while the individuals from Megalofonou, Damalas & De Metrio (2009) and from this work, originated principally as by-catch from commercial fisheries, such tuna and swordfish longline.

Overall, Mediterranean and adjacent North-eastern Atlantic BS displayed a complex geographical structure in which weak but significant differences proved that a certain degree of population connectivity across the Strait of Gibraltar occurred. These results are in contrast with those obtained by tagging data in the past (Kohler, Casey & Turner, 1998; Kohler et al., 2002; Kohler & Turner, 2008; Poisson et al., 2015). Similar findings of genetic differences were observed in other shark species, more related to a benthic environment, such the small-spotted catshark, Scyliorhinus canicula, and the velvet belly lanternshark, Etmopterus spinax (Gubili et al., 2014; Gubili et al., 2016; Kousteni et al., 2015). The reported evidence of genetic structure in the blue shark analyzed in this study are associated with geographical differences in sex-ratio and size. Our results suggest BS in the NE Atlantic and the Mediterranean are not panmictic. There is still no direct observations of mating events take place in the Eastern Mediterranean, but the biological data analysis results support the Eastern Mediterranean as an important nursery area for this species (Megalofonou, Damalas & De Metrio, 2009). Such microevolutionary pattern of differentiation of Mediterranean and North-eastern Atlantic BS prompt the need for a deeper population genetic analysis on the same population samples with more powerful markers for investigating potential subtle structure of BS populations (e.g., microsatellites or SNPs) to provide robust data on BS population structure that are of priority for the BS stock management. High genetic diversity values are usually related to large population size (Frankham, 1996), and the high genetic diversity showed by both Mediterranean and North–eastern Atlantic BS at the two mitochondrial makers advocates in favour of a large size of these populations. Mediterranean and North–eastern Atlantic BS showed higher Cytb gene polymorphism than Pacific BS (Mediterranean and North–eastern Atlantic: h = 0.777–0.814; π = 0.002–0.004; Pacific: h = 0.517–0.768; π = 0.0007–0.0011, Li et al., 2016).

Based on nuclear markers, similar values of observed heterozygosity were detected between Pacific and North Atlantic BS (Sampaio da Costa, 2013; King et al., 2015; Veríssimo et al., 2017). High genetic diversity in abundant species is likely due to a combination of demographic factors, such as local population sizes, fast generation times and high rates of gene flow with other populations (Hague & Routman, 2016). The high genetic diversity shown by Mediterranean and North-eastern Atlantic BS could be a consequence of the short time elapsed, in proportion to the relative generation time, since the population started to suffer overexploitation. In fact, the abundance of the Mediterranean BS has declined by ∼78–90% over the past 30 years (Ferretti et al., 2008), approximately corresponding to three generations; the BS generation time was estimated at 8.2 and 9.8 years for South African and North Atlantic populations, respectively (Cortés et al., 2015). Furthermore, biological characters such as the large size of litters, the low nucleotide substitution rate compared to other vertebrates (Martin, Naylor & Palumbi, 1992), the high potential of migration and the high gene flow between geographical distant populations, may have affected the relationship between genetic diversity and population size, masking the sudden potential population bottleneck of the last three decades, without genetic erosion.

Otherwise, the mismatch distributions of the different macro areas appear to be slightly skewed unimodal, related to a recent bottleneck or a sudden population expansion (Fig. S2), and given the Bayesian skyline plots (Fig. 3), there is overwhelming evidence that the Mediterranean and North East Atlantic populations have undergone a constant population expansion during the last 400–200 Kya, especially within the Mediterranean samples.

The data we show here represent a novelty for the knowledge of Mediterranean blue shark, and our findings highlight the importance of the Mediterranean Sea as nursery area for this species, with direct implication to specific conservation measures for the species.

This work sheds new light on the understudied BS of the Mediterranean Sea, and emphasizes the need of conducting further population genetic surveys on this population. With ongoing efforts, (i.e., https://fishreg.jrc.ec.europa.eu/web/medbluesgen/) a greater understanding of the genetic diversity, spatial population structure and gene flow in this species will be achieved, which will enable us to devise more effective strategies for the management of this increasingly exploited ocean predator.

Supplemental Information

Figure S1 Cytochrome-b (Cytb) and Control Region (CR) Maximum Likelihood Haplotype Network of Mediterranean and North Atlantic Blue Shark populations detailed per sampling locations

Click here for additional data file.

Figure S2 Mismatch Distribution for the four different blue sharks geographical groups for both Cytochrome-b (Cytb) and Control Region (CR) mitochondrial markers

Click here for additional data file.

Figure S3 Cytochrome-b (Cytb), A, and Control Region (CR), B, divergence time analysis of the total BS dataset

Node ages on the node labels. Posterior probability on the branches. 95% HPD bars showed for nodes with Posterior > 0.5.

Click here for additional data file.

Table S1 Collected BS individuals categorized by sex and size across sampling areas and regions

M, males; F, females; na, unsexed; J, juvenile (TL ≤ 120 cm); Y, young (TL = 120–180 cm); L, large (TL ≥ 180 cm). NATL: North-eastern Atlantic; WMED: Western Mediterranean; EMED: Eastern Mediterranean.

Click here for additional data file.

Table S2 Cytb, A, and control region, B, genetic diversity data from the eight sub-populations

South North East Atlantic, SNEATL; North North East Atlantic, NNEATL; West-Mediterranean/Balearic, BALE; South Tyrrhenian Sea, TYRR; Calabria Ionian Sea, IONI; Central Adriatic Sea, CADR; Aegean Sea & Eastern Ionian Greece.

Click here for additional data file.

Table S3 Analysis of molecular variance (AMOVA) of Cytochrome b (Cytb) and Control Region (CR) of the Juvenile and Immature Mediterranean and North-eastern Atlantic Blue Sharks (Prionace glauca)

Click here for additional data file.

Table S4 Pairwise Φst values (below the diagonal) and associated p-values (above the diagonal) among the juvenile and immature blue sharks of the three geographical areas estimated at the two mitochondrial markers (Cytochrome b, Cytb; Control Region, CR)

*Values that resulted not significant after the Bonferroni correction for multiple tests (a-level of significance after Bonferonni correction: p = 0.0166).

Click here for additional data file.

Supplemental Information 1 Control region alignment. The sequences will be deposited in the GenBank upon acceptance of the paper

Click here for additional data file.

Supplemental Information 2 Cytb alignment. The sequences will be deposited in the GenBank upon acceptance of the paper

Click here for additional data file.

We are grateful to all MEDBLUESGEN partners that contributed to the fulfilment of project, including both scientific staff and administrative colleagues. We thank Clara Hugon for the valuable technical support.

Additional Information and Declarations

Competing Interests

Author Contributions

Data Availability

The authors declare there are no competing interests.

Agostino Leone and Ilenia Urso conceived and designed the experiments, performed the experiments, analyzed the data, contributed reagents/materials/analysis tools, wrote the paper, prepared figures and/or tables, reviewed drafts of the paper.

Dimitrios Damalas, Jann Martinsohn, Antonella Zanzi and Stefano Mariani analyzed the data, contributed reagents/materials/analysis tools, wrote the paper, reviewed drafts of the paper.

Emilio Sperone, Primo Micarelli, Fulvio Garibaldi, Persefoni Megalofonou, Luca Bargelloni, Rafaella Franch, David Macias, Paulo Prodöhl, Séan Fitzpatrick and Marco Stagioni contributed reagents/materials/analysis tools, reviewed drafts of the paper.

Fausto Tinti and Alessia Cariani conceived and designed the experiments, analyzed the data, contributed reagents/materials/analysis tools, wrote the paper, reviewed drafts of the paper.

The following information was supplied regarding data availability:

Sequences are available in Genbank with related biological data under accession numbers MG515900–MG516106 and MG545732–MG545901 for Cytb and control region, respectively, and raw data is presented in the Supplemental Files.

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
