# Peer review of "Genetic differentiation and phylogeography of Mediterranean-North Eastern Atlantic blue shark (Prionace glauca, L. 1758) using mitochondrial DNA: panmixia or complex stock structure?"

_PeerJ, doi:10.7717/peerj.4112_

## Round 0.1 · original submission · Minor Revisions

Thank you for submitting your manuscript to PeerJ.

The two reviewers have made suggestions which we feel would improve your manuscript, and I encourage you to consider all these comments and make an appropriate revision of your manuscript.

When submitting your revised paper, please include a separate document uploaded as "Response to Reviewers" that carefully addresses the issues raised in the below comments, point by point. You should also include a suitable rebuttal to any specific request for a change that has not been made.

The reviewers' comments are below.

Please carefully address the issues raised in the comments, such as:
1. Because PeerJ does not offer copyediting, please ensure that your revision meets the English language standards. Lines 12-16, 161-162, 258-264, 294-296, and the overall wordiness of the manuscript (e.g., in lines 258-259 “Based on these results, there seems to be no compelling reason to argue that the Mediterranean BS are fully panmictic with the BS inhabiting the adjacent North-eastern Atlantic” can be changed to “Our results suggest BS in the NE Atlantic and the Mediterranean are not panmictic.”
2. Complexity in structure between the Mediterranean and adjacent North-eastern Atlantic BS, by using other studies with similar distribution that will assist you in justifying the weak structure between the Med and the Atlantic. (ref#2)
3. Reshaping the intro following suggestions of ref#1

·

Basic reporting

No comment

Experimental design

No Comment

Validity of the findings

No comment

Additional comments

The authors use mtDNA sequencing to examine population structure between Mediterranean and Northeast Atlantic blue shark populations. The paper is interesting and will make a nice contribution to the literature. I have made comments and suggestions below.

General comments

Given the proximity of the populations, I was surprised the authors found genetic differentiation in their dataset. This is a very interesting result.

In some areas, the authors are a bit too verbose (examples: last sentence of Abstract; lines 76-80; lines 215-219; lines 237-242; lines 254-258; lines 316-330)

I commend the authors for the interactive map of their sampling—this is very cool!

Why were the sequences not combined to a concatenated sequence and the same set of analyses run? Also, why the discrepancy between sample sizes for the two genes?

Specific comments:

Abstract:

I suggest changing “failed in finding evidences of constant migrations” to “failed to find evidence of migration”

I suggest changing “The network results revealed apparently no genetic structure through the Mediterranean-Atlantic seaway, quite the opposite, the Phist AMOVA and pairwise Phist analysis found a significant genetic structure among four geographical groups” to “Although no obvious genetic structure was apparent from the haplotype network, Phist analyses indicated significant genetic structure among four geographical groups.”

Line 3: make “shark” plural and change “spreads” to “is found”
Lines 17-18: change to “on the western side” and “on the eastern side” (change “in” to “on” in each case).
Line 19: delete “and”
Line 55: delete period after Mediterranean
Lines 65-66: authors cite a study stating that almost all BS caught in the Mediterranean were immature, yet in their dataset (Table S1), this is not the case. The authors do not speculate as to why this is. It would be interesting to touch on this in the Discussion if the authors have any insight.
Lines 74-75: In the abstract, the authors state 8 populations—please clarify.
Line 80: change “Expected” to “These”
Lines 116-119: Normally, final concentrations are reported (this is a minor point).
Line 154: change “females outnumbered significantly males” to “females significantly outnumbered males”
Line 178: Which second one? It looks like only the haplotype with 61 individuals is shared by all populations except NNEATL (which is mentioned in lines 176-178).
Lines 231-236: which markers were used in these other studies and how do they relate to data here?
Lines 242-245: Do you mean female philopatry when you say “reproductive movements?” The mtDNA structure found could very well indicate this, although it would help to have corresponding nuclear data to verify. Also, the authors don’t really stress whether or not these sampling locales were nurseries (although they do mention in passing)—I would suggest explicitly stating which sites are known (or suspected) nursery sites—the authors do this for the Mediterranean but not the Atlantic sites. Also, I would suggest analyzing sites without adults—if female philopatry could explain these results, I would expect an even stronger signal when solely analyzing juveniles/immatures from each site.
Lines 258-260: This isn’t necessarily true—they can be panmictic (with males mating across populations) with female philopatry. Based on other studies, I’d be surprised if nuclear data exhibited structure. Following, (lines 266-268)—nuclear data may not necessarily exhibit structure.
Lines 269-283: This paragraph seems out of place. Most of it belongs in the INTRO.
Lines 316-330: Also seems a bit out of place, with a lot of this belonging in the INTRO. Ending with a comment on the fin trade doesn’t fit with this study. Instead, I would suggest stating the novelty of the Mediterranean population based on your results and how specific conservation measures should be implemented for the population (for example, leaving lines 321-325).

Fig 1: This is minor, but I suggest changing “sampling design” to “sampling sites” as some of these sites weren’t designed but were instead from commercial fisheries.
Fig 1: In both the figure (either on the figure itself or in the legend) and also in the text, please include sample sizes of the four regions.
Fig 2: I understand why the authors have multiple colors within certain haplotypes (for example, three green in haplotype 42), but without knowing the sample locales, these are meaningless. Since the authors break this down in the supplemental figure, I would suggest combining the colors for this figure.

Reviewer 2 ·

Basic reporting

The paper is nicely presented, has an adequate number of references and has a sufficient number of tables and figures. I have added minor suggestions for the authors.


Background
“but genetic structure has not been confirmed even at interoceanic distances.” Remove the word “even”. There are plenty of recent papers on BS describing genetic structure on an intraoceanic level.
Results
Remove “apparently” and replace “quite the opposite” with whereas.

Introduction
Line 3: Please use a reference for the previous statement.
Line 40: The authors use the term mating clubs. It might be better to include them into quotation marks.
Line 55: remove the first “.”
Line 71: Add a space before “On the other hand”

Material and Methods
Lines 87-90: Please add the number of individuals per sampling area in case readers will not have to check supplementary material.
Lines 110-111: Why did the authors required to design new primers for the species, since they could follow previous attempts? See Verissimo et al., 2017 for CR primers. Can they justify their decision?
Line 120: replace “annealing at 60ᵒC for 30s” with “annealing at 60ᵒC for 30 s”.
Line 139: Describe the four groupings for the AMOVA analysis.

Results
Line 179: What does the N≥10 mean? Number of individuals? Number of haplotypes? Please clarify.

Discussion and Conclusion
Line 244: Replace “Φsts” with “Φst values”.

Lines 252-253: I am wondering about the role of the Adriatic as a nursery for BS. The authors claim to confirm Megalofonou et al., 2009 suggestions about the matter. I also assume that the authors have probably used samples provided from the same dataset used in that manuscript. Considering the lack of information on sampling dates (I do not remember seeing a table with the actual dates that the samples were caught), have the authors compared the sizes or sex ratios based on time of capture? Do they have other individuals caught in the same period for direct comparison and subsequently prove their argument?

Figures and Tables
Figure 2: What is the point of the additional small figure displaying the size+number of individuals per haplotype, when you have the number of individual in each haplotype? Please remove it, as it does not add anything to the Figure.
Figure 3: You need to add the details for the sampling locations. NNEATL: North North-eastern Atlantic; SNEATL: South North-eastern Atlantic; WMED: Western Mediterranean; EMED: Eastern Mediterranean.

Experimental design

The importance of the sampling area is highlighted throughout the manuscript. The Mediterranean is a very important mating and nursery ground for chondricthyans. The methods are sufficiently described.

However, I was wondering about the sampling dates. Sampling took place from 2003 to 2016. Have the authors checked for temporal differences?

Validity of the findings

Despite the complexity of the results, the authors do a very good job describing and justifying their results. Their findings are robust. and statistically sound. There are a couple of things that need to be clarified.


There is a lack of discussion regarding the sex ratio differences found between the two seas despite the results they reported in Lines 154-164. Do the authors have any suggestions? Moreover, can you justify the differences in male/female sex ration found in the Mediterranean between your results and the ones reported by Megalofonou et al., 2009?

Lines 254-268: You are trying to justify the complexity in structure between the Mediterranean and adjacent North-eastern Atlantic BS. You are doing an excellent job by justifying your data with the results from tagging data (previous studies). Considering the lack of nuclear data, I believe that you can also use studies with similar distribution might partially help you justify this weak structure between the Med and the Atlantic. There are at least three papers (2 on Scyliorhinus canicula and one on Etmopterus spinax) that might be able to help the authors add to their discussion, particularly in the absence of nuclear data.

Additional comments

This is a nice study with sufficient mtDNA data that is well analyzed and well presented. The figures are clear. I feel that the study will be of broad interest to readers of the journal because of the interesting region being sampled. Our understanding of elasmobranch genetic structure is greatly improved by studies such as this. This is especially true in light of recent papers that have presented genetic structure in the Mediterranean.

I think there is the basis for a nice publication on the population structure of Prionace glauca, and I believe that the manuscript could be improved as few things are not presented adequately.

---

## Round 0.2 · accepted · Accept

Thank you for your revisions of the manuscript in response to both referees' suggestions. I have read through the manuscript and am satisfied with your response and revisions, such that I am happy to move the manuscript forward.

Reviewer 2 ·

Basic reporting

The paper is nicely presented and questions/concerns raised by both reviewers have been dealt with by the authors.

Experimental design

The importance of the sampling area is highlighted throughout the manuscript. The methods are sufficiently described. I would like to thank the authors for the extra analyses they performed and replying to all questions raised.

Validity of the findings

My concerns from my previous review have been addressed.

Additional comments

This paper is a nice contribution to the literature, particularly for Mediterranean species.